# Future of Structured Lipids: Enzymatic Synthesis and Their New Applications in Food Systems

**DOI:** 10.3390/foods11162400

**Published:** 2022-08-10

**Authors:** Jun Zhou, Yee-Ying Lee, Yilin Mao, Yong Wang, Zhen Zhang

**Affiliations:** 1JNU-UPM International Joint Laboratory on Plant Oil Processing and Safety, Department of Food Science and Engineering, Jinan University, 601 Huangpu Ave West, Guangzhou 510632, China; 2School of Science, Monash University Malaysia, Bandar Sunway 47500, Selangor, Malaysia; 3Guangdong Joint International Research Centre of Oilseed Biorefinery, Nutrition and Safety, Guangzhou 510632, China

**Keywords:** structured lipids, enzymatic catalysis, food specialty fats, future food

## Abstract

Structured lipids (SLs) refer to a new type of functional lipid obtained by modifying natural triacylglycerol (TAG) through the restructuring of fatty acids, thereby altering the composition, structure, and distribution of fatty acids attached to the glycerol backbones. Due to the unique functional characteristics of SLs (easy to absorb, low in calories, reduced serum TAG, etc.), there is increasing interest in the research and application of SLs. SLs were initially prepared using chemical methods. With the wide application of enzymes in industries and the advantages of enzymatic synthesis (mild reaction conditions, high catalytic efficiency, environmental friendliness, etc.), synthesis of SLs using lipase has aroused great interest. This review summarizes the reaction system of SL production and introduces the enzymatic synthesis and application of some of the latest SLs discussed/developed in recent years, including medium- to long-chain triacylglycerol (MLCT), diacylglycerol (DAG), EPA- and DHA-enriched TAG, human milk fat substitutes, and esterified propoxylated glycerol (EPG). Lastly, several new ways of applying SLs (powdered oil, DAG plastic fat, inert gas spray oil, and emulsion) in the future food industry are also highlighted.

## 1. Introduction

Lipids are among the vital nutrients needed for supporting normal physiological activities and aiding in the maintenance of human health by providing essential fatty acids. However, excessive fat consumption can lead to obesity and related diseases such as diabetes, high blood pressure, hyperlipidemia, hardening of the arteries, and heart disease. At present, obesity and cardiovascular diseases have become health threats worldwide. A variety of fat-substitute products, such as starch, cellulose, protein, pectin, and glucose, have been developed to control excessive fat consumption [1], although they have not been shown to be effective in obese individuals [2]. The lack of self-discipline on strict dietary regimes for dieters may cause weight rebounds [3]. This might be related to the low palatability of low-fat diets. As a result, developing novel oils and fats with weight loss function is of great significance in replacing traditional edible oils and fats.

Structured lipids (SLs) represent a form of oil that has unique functional properties, such as being easily absorbed, low in calories, and low in saturated fat. SLs are obtained by modifying natural triacylglycerol (TAG) through the restructuring of fatty acids, thereby altering the type, content, structure, and/or distribution of fatty acids attached to the glycerol backbone [4]. However, SL products are still in the laboratory preparation stage, and most present reaction systems are not suitable for food production, while the production cost is high. Hence, how to reduce the cost by changing the preparation method and selecting decent large-scale reactors to improve the industrialization and practical application of SLs is still a hot research topic. The SLs can be produced using chemical, enzymatic, and genetic modification methods, among which chemical methods are the primary preparation approach practiced in industries due to their low cost, ease of processing, mass production, etc. However, chemical methods have some disadvantages, such as low specificity, high energy consumption, high reaction temperature, and easy degradation of reactants and products, and the products may exhibit risk factors that are not conducive to consumer health. Therefore, researchers are actively exploring the preparation of SL using enzymatic methods with mild conditions, strong specificity, and low risk factors.

The uniqueness of SLs lies in their structural characteristics, which influence the stomach emptying rate and fat digestion, thus providing the desired physiological attributes that improve the nutritional properties [5]. SLs can reduce the accumulation of fat in the body and improve metabolic disorders such as fat malabsorption, thereby reducing the incidence of obesity and cardiovascular disease, in addition to being beneficial for fetal development. However, using SLs has some drawbacks such as poor oxidative stability and short product shelf-life. Adding synthetic antioxidants can improve the stability of SLs; however, this does not conform to the concept of healthy food. On another note, the oil’s quality and stability can be enhanced by establishing a new and stable food system for SLs that aligns well with its application and indirectly expands the diverse range of available fat and oil products. In the future, SLs could be applied in microcapsule oils, plastic fats, and food emulsions to better meet people’s health needs. Some novel application methods, such as in spray oil stored in an inert gas container, could also improve the oxidative stability.

Most of the previous reviews summarized the specific nutritional functions of some SLs, but there is less understanding of the future applications in food systems for recent SLs. In this regard, this article encompasses a review of the enzymatic production and application of recent SLs, including medium- to long-chain TAG (MLCT), diacylglycerol (DAG), eicosapentaenoic acid (EPA)- and docosahexaenoic acid (DHA)-enriched TAG, and human milk fat substitutes (HMFSs) (Figure 1). New applications of SLs in food systems in future are also discussed.

## 2. Reaction System for SL Production

The modification of fat and oil can be achieved through chemical, enzymatic, and genetic methods. The traditional chemical method for the preparation of SL has unique advantages, but it is not suitable to be used in food systems because of its disadvantages such as the production of dangerous factors at high temperature. Application of the genetic method is limited by the problems of market supervision and consumers’ skeptical attitude toward the products. At present, the enzymatic synthesis of SLs shows great advantages, e.g., mild reaction conditions, avoidance of the production of risk factors, and the generation of immobilized enzymes, which greatly reduce the cost of enzymatic production [4]. Table 1 shows the advantages and disadvantages of the three methods for preparing SLs.

Enzymatic synthesis takes place in a solvent-free system under relatively moderate reaction conditions. This process has the advantages of being nontoxic (no hazardous organic solvents) and easy to operate, with rapid and simple purification steps, using enzymes that can be reused for numerous cycles, which is conducive to the manufacturing of high-value products [6]. As a result, the solvent-free approach has broad development prospects for the industrial-scale enzymatic synthesis of SL.

For oils with a high viscosity coefficient, it is preferable for the reaction to be performed in an organic solvent solution. This is because the presence of organic solvents can significantly reduce the viscosity of the reaction system and increase the solubility of the oil, allowing the reaction to be carried out at a lower temperature and, thus, consuming less energy, as well as reducing the occurrence of acyl migration and hydrolysis reactions [7]. The lipase is suspended in an organic solvent and may be filtered and separated rapidly when the reaction has completed. Nevertheless, purification of the end product can be a challenging task. Most researchers used organic solvents during SL production in order to improve the substrate solubility; however, this is not suitable for food applications because of the solvent residues. However, the industrial production of edible-grade functional oils has begun to expand the use of safer, continuous, solvent-free systems.

The enzyme reactor is the core device for an enzyme-catalyzed reaction. It provides an appropriate setting for the reaction to occur under optimal enzyme-catalyzed reaction conditions, allowing the substrate to be converted into products to the greatest extent possible. Despite the fact that several studies have shown that enzymatic transesterification to synthesize SL has apparent benefits [17,18,19], production at an industrial scale is still out of favor. Successful industrial-scale production requires to be not only economically feasible, but also strongly supported by pilot plant technology. The two most popular reactors used in enzymatic transesterification are stirred tank reactor and packed-bed reactor (PBR) (Figure 2), with the former being more commonly studied. SLs with enriched DHA and ARA at *sn*2 were successfully produced by reacting oil substrates in a solvent-free stirred tank system [20]. Lee et al. [21] discovered that enzymatic transesterification using palm oil as the raw material produced 60% palm oil-based MLCT in a pilot-scale test. Irrespective of the scale of the reactor, stirred tanks have intrinsic benefits such as inexpensive equipment costs, a simple device, complete reaction, and controllable reaction conditions. However, the immobilized enzyme must be retrieved periodically throughout the reaction, making the conditions tedious. Furthermore, the stirring paddle has a high shear force, making it easier to shatter and break enzyme particles, which inhibits enzyme activity. PBR has been successfully used as a large-scale reactor in industrial-scale application studies. PBR is also the most extensively employed reactor for heterogeneous catalytic systems involving solid–liquid interfaces. PBR has shown its unrivalled advantages in synthesizing SL, particularly in industrial settings. The scale-up of PBR for enzymatic transesterification of edible waste oil is also feasible, as confirmed by Halim et al. [22]. PBR offers a simple design structure, high efficiency, ease of operation, better stability of the immobilized enzymes, mild reaction conditions, nontoxicity, and high recyclability [23].

Traditional enzyme reactors employ mechanical stirring or liquid flow for mass transfer, but the reaction efficiency may be affected by the reaction system’s high viscosity or the non-miscibility of reactants [24,25]. However, vigorous stirring to increase mechanical power substantially increases the risk of rupturing the enzyme carrier, thereby leading to enzyme inactivation. The concept of a bubble column reactor (Figure 2) was initially proposed by Hilterhaus et al. [26], in which an immobilized enzyme catalyzed the esterification of lauric acid and polyglycerol in a solvent-free system. The high-viscosity reaction system was gently and effectively agitated by bubbles in BCR using nitrogen. This stirring method has excellent heat and mass transfer efficiency, low destruction of enzyme support, high catalyst durability, and particular feasibility for heterogeneous systems and exothermic reactions [27]. Table 2 summarizes the reactors used for the enzymatic synthesis of SLs in recent years.

## 3. Enzymatic Preparation and Application of SLs

### 3.1. Medium—To Long-Chain Triacylglycerol (MLCT)

A new type of SL, MLCT, composed of fatty acids of medium- and long-chain lengths randomly positioned on the same glycerol backbone, was synthesized to improve the safety and efficacy of medium-chain TAG (MCT) fat emulsions and avoid the shortcomings of MCT/long-chain TAG (LCT) physically mixed fat emulsions. Due to its simultaneous supplies of medium-chain fatty acids (MCFAs; fatty acids with 6–12 carbon atoms in the carbon chain) and long-chain fatty acids (LCFAs; fatty acids with more than 12 carbon atoms in the carbon chain), MLCT maintains the advantages of MCT in terms of easy digestion and provision of a rapid source of energy, while the LCFAs contained in MLCT supplement essential fatty acids. Compared to the physical mixture, MLCT exhibits superior fatty-acid metabolism, allowing it to be cleared from the bloodstream more quickly while improving nitrogen balance without reducing liver function. It has been reported that MLCT can be used to control obesity, fat malabsorption, and other metabolic disorders, while also improving insulin resistance to regulate diabetes, as it is well regarded as a healthy cooking oil [21,37]. Nevertheless, MLCT is rarely found in nature in a high concentration, necessitating its synthesis.

MLCT is synthesized mainly through one-step or multistep enzyme-catalyzed reactions with lipase. Depending on the reaction substrate, one-step enzymatic catalysis includes acidolysis, interesterification, and esterification [21] (Figure 3). Multistep enzymatic catalysis combines two types of one-step reactions (esterification and acidolysis, esterification and acidolysis, etc.) to produce MLCT. Acidolysis, the reaction between triacylglycerol and free fatty acids, is the most widely used method for producing MLCT [37,38]. Usually, capric acid, caprylic acid, and lauric acid are utilized as the source of MCFAs, whereas corn oil, mango kernel fat, avocado oil, soybean oil, *Echium* seed oil, canola oil, corn oil, olive oil, and microbial oil are used to provide LCFAs. In esterification, the substrates are generally glycerol and fatty acids. Esterification provides a higher concentration of MLCT as compared to transesterification and acidolysis due to the higher purity of MCFAs and LCFAs. Nevertheless, esterification has some drawbacks in that it possesses no natural antioxidants and is costly due to the expensive fatty acids and glycerol. Interesterification is the reaction involving the exchange of two acyl groups between two ester molecules or TAGs. Most of the current studies prepared MLCTs by inter-esterifying oils rich in MCFAs (palm kernel oil, coconut oil, *Cinnamomum camphora* seed oil, etc.) with common vegetable oils such as rapeseed oil, soybean oil, palm oil, and *Camellia* oil. This process is preferred because it generates a small amount of byproducts such as free fatty acid, MAG, and DAG. Polyunsaturated fatty acids (PUFAs) are usually used as the substrate in a two-step reaction. Some studies were aimed at esterifying glycerol and PUFAs to generate tri-PUFAs, followed by acidolysis of the tri-PUFAs and MCFAs to produce MLM SLs [39]. Table 3 summarizes the enzymatic synthesis of MLCT in recent years.

Fat and oil are required to provide energy to the body and maintain human health. Moreover, food products are bestowed with unique flavors attributable to the features of different fatty acids present in oil. Hence, fats should not be entirely removed from foods in order to fulfill both the health and the technological aspects. Nevertheless, consuming foods with high-fat content can be detrimental to health. MLCT offers great potential as a replacement for traditional oils, especially in high-fat diets, to manage/prevent weight gain and body fat accumulation. Several current studies have demonstrated the potential of MLCT as an ingredient in edible oils, energy bars, margarine, shortening, and beverages. Recently, MLCT has been sought after as valuable functional oil for human milk fat substitute due to the close similarity of its triacylglycerol and fatty-acid composition with breast milk, particularly the MLL species. MLCT has become a research hotspot, and it is widely employed in the food, cosmetic, and pharmaceutical sectors as a functional oil [39,48].

### 3.2. Diacylglycerol (DAG)

DAG is a class of SL in which one of the fatty-acid chains on the glycerol backbone is replaced by a hydroxyl [49]. It can be found in trace amounts in natural oils and fats [50,51]. The two DAG isomers, namely, 1,3-DAG and 1,2(2,3)-DAG, are named according to the positions of hydroxyl groups at *sn*2 and *sn*3 (1), with ratios of about 7:3 and 6:4, respectively [52]. This ratio may also approach 1:1 due to the occurrence of acyl migration during deodorization and storage. Because of its symmetrical structure, 1,3-DAG is more thermodynamically stable in terms of physical and chemical characteristics. Generally, the melting point of 1,3-DAG is about 10 °C higher than that of TAG with the same fatty-acid composition, while the melting point of 1,2-DAG is similar to that of TAG with the same fatty-acid composition. Furthermore, 1,2-DAG tends to form α and β’ crystals, while 1,3-DAG is more symmetrical and tends to form stable β crystals [53]. Generally speaking, TAG products mainly focus on the type and content of fatty acids, while the development of DAG products pays attention to the nutritional properties of DAG arising from its unique structure. DAG is beneficial to health mainly because of two aspects: inhibiting the increase in postprandial serum TAG and suppressing the accumulation of body fat, mainly because of the difference in the metabolic modes compared to TAG [54]. Most 1,3-DAG is hydrolyzed into glycerol and free fatty acids, which are transported to the liver for energy by β-oxidation. Part of the intermediate product 1-MAG, which is not entirely hydrolyzed, is absorbed by small intestinal epithelial cells and resynthesized into new TAG. Nevertheless, this process is significantly less efficient than direct intake of TAG; hence, DAG plays a significant role in the prevention and treatment of obesity [55].

DAG is traditionally produced via chemical catalytic glycerolysis, which is conducted at high temperatures [56]. With the development of enzymes for food application in the industry, commercial lipases have been widely utilized to prepare DAG via partial hydrolysis, glycerolysis, esterification, and transesterification [52] (Figure 3). TAG may be over-hydrolyzed to MAG and fatty acids instead of DAG during hydrolysis, which limits the use of partial hydrolysis process for DAG production. As for glycerolysis, one of the substrates, i.e., glycerol, is easily adsorbed on the surface of the enzyme carrier, forming a hydrophilic layer that restricts the interaction between lipase and oil, limiting the reaction [57]. To ensure that the reaction proceeds forward, the esterification procedure must continually eliminate the water generated during the reaction. Moreover, glycerol and oil phases that are immiscible in glycerolysis and esterification reduce the mass transfer of the reaction process [58]. There are a variety of approaches to solve this challenge, including the introduction of organic solvents of different polarities to enhance the glycerolysis process by promoting the uniform distribution of two phases in the reaction system. Surfactants/ionic liquids, supersonic assistance, supercritical fluids, or suitable reactors can also increase the contact surface of reaction substrates with enzymes and further improve the yield [59,60,61,62]. However, these technologies are too costly and energy-intensive for industrial manufacturing, making them unsuitable for large-scale production. Table 4 summarizes the enzymatic synthesis of DAG in recent years.

Substituting DAG for normal edible oil can not only affect appetite but also inhibit weight gain due to DAG’s unique metabolic route in the human body [54,72]. As a result, DAG may be utilized to formulate foods with weight loss function that are nutritious and healthy. Bread baked with DAG-rich fat was reported to be fluffier, taste better, and be easily molded [73]. Lecithin, DAG, and other additives have been developed as promoters that may be dissolved in instant drinks to accelerate the dissolution of solids and provide products with a smooth mouthfeel. Dough prepared with DAG shortening has good oil-retaining properties, is easy to cut, and has good stability [74].

The presence of a hydroxyl group in the DAG molecule endows DAG with certain hydrophilicity. In terms of food applications, this feature may influence the retention of some flavor substances, hydrolysis during cooking, selection of food emulsifiers, and solubility of some functional components in foods. DAG can also be used in the pharmaceutical industry [75]. DAG has the function of inhibiting bile-acid secretion; it may be manufactured into tablets, powders, and capsules, as well as suppositories and injections, for the prevention and treatment of diarrhea. In addition, DAG can be mixed directly with medications, acting as a slow-release agent to regulate drug release and enhance the functionality of other active ingredients in the medication. DAG is also a synthetic raw material in the chemical industry, and it can also be utilized in skin care products to increase the skin’s water-holding capacity [76]. DAG containing special fatty acids also has a good repair effect on frostbitten skin.

Alkylglycerol (AKG) belongs to the ether lipid family. It is a hydroxyl compound of the glycerin molecule, with the *sn*1 or *sn*3 bit connected to the hydrocarbon group (alkyl or alkenyl) via an ether bond; therefore, it is also known as hydrocarbon ether glycerin, including alkyl ether glycerin and alkylene glycerin. Alkylglycerols are acylated into alkyl glycerides, whereby the hydroxyl group of the glycerol molecule *sn*1 is linked to the hydrocarbon group via an ether bond, while the *sn*2 and *sn*3 hydroxyl groups form DAG with two molecular fatty acids, which can be regarded as derivatives of DAG (Figure 1). Due to the different length and saturation of the alkyl group, there are many kinds of AKGs such as batyl alcohol and selachyl alcohol. Research has shown that AKG regulates the transformation of body fat in infants, thereby preventing obesity [77].

### 3.3. EPA and DHA-Enriched TAG

EPA and DHA are polyunsaturated fatty acids that are abundant in marine animals and plants; for example, fish oil contains high levels of EPA and DHA. The concentration of EPA and DHA in fish oil products on the market is around 18% and 12%, respectively. Several studies have shown that EPA and DHA are important for fetal development, including retinal, neuronal, and immune development. EPA and DHA may also be associated with cardiovascular function as anti-inflammation and anti-coagulant agents in managing coronary and peripheral artery disease. In addition, they have been shown to be effective in weight management and cognitive function improvement for those with mild Alzheimer’s disease [78,79]. Consumers are becoming more health-conscious, and there is a greater demand for EPA and DHA products. Hence, developing products with high concentrations of EPA and DHA is of top priority.

The enrichment of EPA and DHA in fats and oils can be achieved via physical, chemical, or enzymatic methods such as low-temperature solvent crystallization, supercritical fluid extraction, urea complexation, and silver nitrate complexation [80,81,82,83,84,85]. Via these techniques, EPA and DHA in fish oil are first hydrolyzed or trans-esterified into their free form or ethyl ester before enrichment. Because of its moderate reaction conditions, remarkable specificity, high catalytic efficiency, low energy consumption, and excellent product safety, the enzymatic approach has become a preferred method for enriching EPA and DHA in fish oil in recent years [86,87]. Table 5 summarizes the synthesis of EPA- and DHA-enriched TAG in recent years.

The primary component of natural fish oil is a TAG of mixed fatty acids. Via the positional or acyl selectivity of lipase during the lipid modification process, i.e., hydrolysis [86], transesterification, and esterification, fish oil with high concentrations of EPA and DHA can be obtained (Figure 3). The *cis*-carbon double-bond structure of EPA and DHA results in a highly coiled fatty-acid chain, leading to close proximity of the methyl group with the ester bond in the glycerol ester molecule, thus resulting in the active site of the lipase being difficult to be accessed by EPA, as well as an ester bond formed between DHA and glycerol. In contrast, the straight-chain structure of saturated and unsaturated fatty acids (low degree of unsaturation) does not suffer from steric hindrance or has low steric hindrance, making them more susceptible to hydrolysis. Under the activity of enzymes, the saturated and unsaturated fatty acids (low degree of unsaturation) on glycerol esters are hydrolyzed, thereby increasing the EPA and DHA contents. Transesterification refers to the lipase-catalyzed acyl exchange reaction between the glyceride and free fatty acids (EPA and DHA) or alcohol or EPA and DHA esters, with the intention of enhancing EPA and DHA. Alcoholysis, acidolysis, and transesterification are the three types of transesterification processes [100,101]. To accomplish the goal of EPA and DHA enrichment, the enzymatically catalyzed esterification process selectively catalyzes EPA and DHA into their free fatty-acid form, causing them to react with glycerol and further enrich the glycerol esters. This process fully hydrolyzes fish oil glyceride into free fatty acids, concentrates and enriches them to high concentrations of free EPA and DHA, or saponifies fish oil glyceride ethyl ester to obtain free fatty acids, before esterifying them with glycerol under lipase catalysis [102]. The lymphatic recovery of EPA and DHA given as TAG was faster and more efficient than corresponding ethyl esters and free acids, particularly shortly after administration. The limited related reports call for more studies exploring the enrichment of fish oil products with EPA and DHA in the form of TAG. Thus, economical and efficient enzymatic processes to prepare EPA- and DHA-enriched TAG should be investigated in the future.

### 3.4. Human Milk Fat Substitutes

Human milk fat is considered the best source of dietary fat for infants, providing 50% of the dietary energy demand for infants [103], in addition to being an important source of essential fatty acids, fat-soluble vitamins, and bioactive compounds [104]. Human milk fat TAG’s composition consists mainly of linoleic acid, palmitic acid, and oleic acid. According to the position distribution, over 70% of palmitic acid in human milk fat is found in the central position. UPU-type TAGs (unsaturated fatty acids at *sn*1,3 positions and palmitic acid at the *sn*2 position) are typical TAGs in HMF. Hence, human milk fat is abundant in 1-oleoyl-2-palmitoyl-3-oleoylglycerol (OPO) and 1-oleoyl-2-palmitoyl-3-linoleoylglycerol (OPL) [105]. The milk fat TAG structure and the distribution of fatty acids significantly impact the nutrition absorption in infants. Infants more readily absorb fats containing palmitic acid at the *sn*2 position under the age of 12 months with lower total soaps, palmitate soaps, and calcium [106]. The addition of OPO and OPL into infant formula can, thus, be more easily absorbed by the human body [107], suggesting a new research direction for human milk fat substitute formulation.

There are three main types of enzymatic reactions, i.e., acidolysis, transesterification, and multistep enzymatic catalysis, to synthesize OPO and OPL (Figure 3). The most direct method is the acidolysis of an oil rich in palmitic acid at the *sn*2 position with linoleic acid and/or oleic acid, catalyzed by widely used 1,3-specific lipases. Transesterification is a reaction between a TAG and TAG or an ester, usually occurring between oils rich in *sn*2 palmitoyl and *sn*1,3 oleoyl. Due to its simplicity, a one-step enzymatic method has been used in many studies, but it also has some disadvantages, such as the difficulty in converting intermediate DAG to HMFS and the complexity of purification due to the existence of byproducts [108,109]. To overcome these shortcomings, researchers have studied multistep enzymatic catalytic reactions (alcoholysis and esterification, acidolysis and acidolysis, transesterification and acidolysis, etc.). Firstly, the natural oil rich in *sn*2 palmitoyl is subjected to alcoholysis by 1,3-specific lipase, yielding *sn*2 palmitoyl MAG. Then, the MAG is esterified with linoleic and oleic acid to form the OPO/OPL [110]. Table 6 summarizes the synthesis of OPO/OPL in recent years.

The majority of studies focused on the OPO synthesis process. In contrast to OPO, OPL contains linoleic acid, which is an essential fatty acid that can be elongated and desaturated to form other omega-6 polyunsaturated fatty acids, such as γ-linolenic acid and arachidonic acid, which are vital for brain and retinal development during pregnancy [114]. Furthermore, LA is an indispensable structural component of certain skin ceramides, critical for maintaining the water barrier of the skin epidermis. Most vegetable oils do not contain OPL and the composition of TAG in animal fats differs from that of human milk fat; hence, it is critical to investigate OPL synthesis. Future research studies should focus on the development of OPO and OPL through enzymatic synthesis using plant-based oil resources as a healthy alternative to the currently used animal fats, which has a high cholesterol content. In addition, the enzymatic cost can be further reduced through choosing the economic lipase or using the reusable immobilized lipases.

**Figure 3 foods-11-02400-f003:**
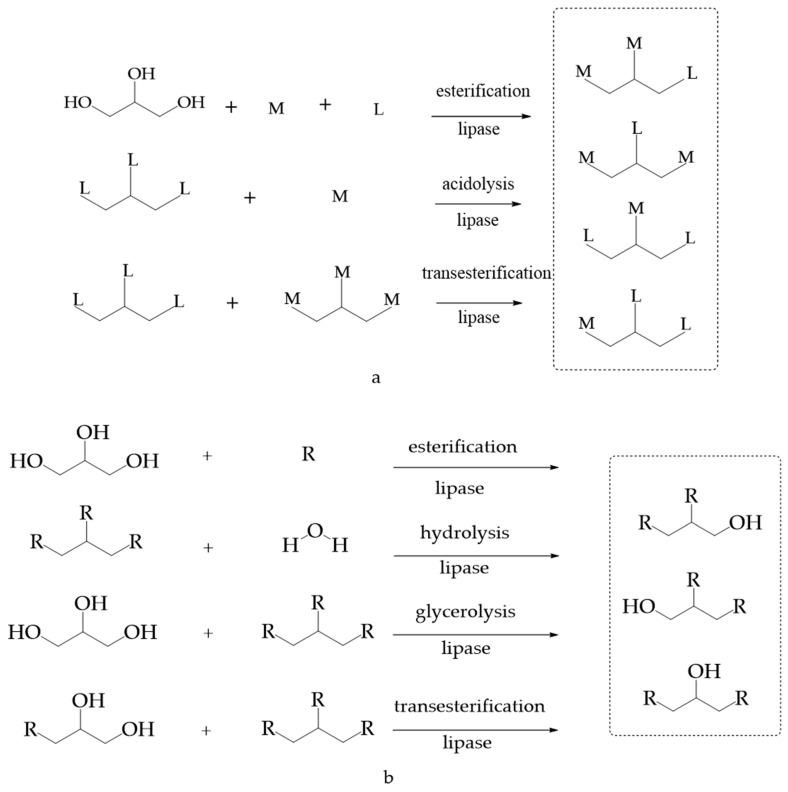
Approaches for enzymatic modification to obtain SLs. (**a**) The main enzymatic synthesis methods of medium-long chain triacylglycerol (MLCT) [6,41,42]. (**b**) The main enzymatic synthesis methods of diacylglycerol (DAG) [53,63,65]. (**c**) The main enzymatic synthesis methods of EPA- and DHA-enriched TAG [86,88,90,92]. (**d**) The main enzymatic synthesis methods of human milk fat substitutes [108,109]. (R, M, L, C_18_C_34_O_2_, C_16_H_32_O_2_, C_18_H_32_O_2_ represent fatty acids, medium-chain fatty acids, long-chain fatty acids, oleic acid, palmitic acid, linoleic acid, respectively).

### 3.5. Esterified Propoxylated Glycerol (EPG)

Esterified propoxylated glycerol (EPG) is an alternative fat consisting of acetyl epoxides and polyols. It is first generated by conjugating glycerol and propylene oxide under alkaline conditions, followed by esterification with fatty acid. Its propoxy group is bound between the glycerol backbone and the fatty acid (Figure 4). Studies have shown that EPG is nontoxic to humans in the context of expected levels of intake and does not accumulate in humans after ingestion [115]; it has been certified by the US FDA as generally recognized as safe (GRAS) [116]. EPG has both the rheological properties of liquid oil and the texture properties of solid oil, along with good palatability. The state of the two phases in EPG depends on the average propoxylation degree (the number of propylene glycol multimers between the glycerol backbone and fatty acid) and the fatty-acid composition, whereby a higher average propoxylation degree results in a more liquid EPG [115].

The similarity of the structural characteristics of EPG and TAG can be used to predict their physicochemical properties. The intermediate metabolite 2-MAG produced after digestion of TAG is finally converted into TAG chylomicrons that increase the metabolic burden [117], while the propylene glycol group that exists between the glycerol backbone and acyl fatty acids in EPG is separated from the free fatty acids after digestion. The methyl group blocks lipase from accessing the ester bond [115], preventing the formation of 1-MAG, thereby avoiding the generation of TAG chylomicrons. For production applications, EPGs with a high average degree of propoxylation are less sensitive to lipases and, thus, may have more beneficial effects in human metabolism. Esterified propoxyglycerol, based on non-GMO vegetable oils (primarily canola), was invented to reduce fat; it is utilized in the preparation of candy, chocolate bars, and baked goods with reduced calories, as well as other applications, e.g., snack foods (chips, corn-flavored snacks, and chicken nuggets), vegetable protein products, beverages, coffee, tea, and dairy analogs. It can replace 50–85% of fat in most applications, reducing calories by up to 45%. At present, there are fewer reports on EPG synthesis and applications. In the future, EPG can be used in many fields as a new lipid substitute to traditional oils. Therefore, the enzymatic synthesis, characteristics, and application properties of EPG used in different food systems deserve to be studied by researchers in future work.

## 4. Future Applications of SLs in Food Systems

Most SLs are made up of unsaturated fatty acids that get easily oxidized. As a result, the oxidation issue has emerged as the most significant challenge for their application and storage. Meanwhile, many dietary supplements have a limited absorption rate when consumed directly. The absorption efficacy of certain fat-soluble functional elements by the human body via novel food systems is an important fundamental aspect for food scientists, which remains a research challenge. In recent years, SLs as functional food ingredients with biological activity and health-promoting functions have steadily attracted increasing attention. Henceforth, this section briefly introduces several potential applications of structural oils.

### 4.1. Powdered Oil

Microencapsulated oil, commonly known as powdered oil, is obtained by wrapping a core material with a water-soluble wall material. It is an oil-in-water (O/W) product made using vegetable oil, maize syrup, high-quality protein, stabilizer, emulsifier, and other auxiliary components using high-tech microcapsule technology developed from the concepts of bioengineering and food engineering. At present, the methods for preparing microencapsulated oil include freeze-drying, spray-drying, complex coacervation, and electrostatic spraying.

In general, the major difficulty in producing healthy oils for food applications is associated with the susceptibility of oils to oxidation. Light, heat, oxygen, etc. can lead to rapid oxidative degradation of oily food. At the same time, spoilage caused by the oxidation of unsaturated fatty acids affects their subsequent application in the food system; for example, the oxidation of fish oil affects the product quality, flavor, color, and shelf-life. The microencapsulation of oils can effectively solve the above problems through the preparation of stable powders with excellent performance. Due to the presence of the wall material layer, which isolates the oil from the external environment, the shelf-stability of the microencapsulated oil product is enhanced. Because the oil particles are embedded in the capsule wall, the product possesses a variety of unique features. Attributed to their superior solubility, fluidity, and small particle size, powdered oil products are widely used in the production of infant dairy products, formula milk powder, milk-containing drinks, cakes, cold food, drinks, pasta, candy, meat products, etc., thereby facilitating their digestion and absorption in the human digestive system [118,119,120]. At present, there are few reports on the microencapsulation of SLs, with more studies focused on the microencapsulation of traditional oils. In the future, SLs, which are easily oxidized and unstable, can be prepared into microcapsule oils to improve their availability. Table 7 summarizes the preparation of powdered oil in recent years.

### 4.2. DAG-Based Plastic Fat

Baking oil is a kind of specialty oil used in the food industry. Emulsification and kneading are used to create a water-in-oil (W/O) emulsion system from base oil. The oil level is typically about 80%, and the quality of the oil directly influences the color, flavor, and taste of baked goods. DAG made from typical unsaturated vegetable oils is easily oxidized; however, DAG made from high-melting-point vegetable oils such as palm oil have higher durability. DAG also has a greater melting point and broader plasticity than TAG with the same fatty-acid makeup, and it may be used to replace high-melting-point solid fat in specialty oils. DAG has been used in baking oil, mainly due to its amphiphilicity, lower interfacial tension between oil and water, and ease of emulsification. Although it is utilized as a food additive or a crystallization inhibitor [53], there are few accounts of DAG being employed as the primary stearin base material. The crisping function of baking oil is based on solid-phase oil. DAG’s high melting point provides a solid foundation for its polymeric skeleton in a W/O emulsion. The universality of DAG and ordinary edible oil makes it practicable to convert DAG into plastic fat products with beneficial functional properties to replace hydrogenated and saturated fat.

The majority of DAG-rich oil research focused on improving the process, while numerous studies were aimed at improving the process optimization and physiological functions; however, there is a scarcity of studies exploring the physical and chemical properties of DAG-enriched oils, as well as the impact of processing conditions on the DAG molecular structure [54,59,129,130]. There are limited studies looking at the influence of the molecular structure, physicochemical characteristics, crystallization behavior, and interaction mechanism of high-melting-point DAG on W/O emulsion systems. As a result, understanding the crystallization and emulsification mechanisms of DAG-enriched oils and fats is critical for developing a baking oil system with good stability and operability, excellent nutrition profile, and low degree of saturation, which is critical for the food industry’s transformation and development.

### 4.3. Inert Gas Spray Oil

Using traditional cooking oil enables us to evenly and quantitatively control the oil usage, whereas oil spray allows the edible oil to be evenly spread out, thereby aiding in controlling and reducing the intake of dietary oil, which is beneficial to health. The use of oil spray originated from a Western cooking habit; currently, the demand for edible spray oil in the world is gradually increasing. The design of edible spray oil allows the application of a controllable quantity of oil. As the packaging of edible spray oil is generally not too large with good sealing performance, it is convenient to carry around. Edible spray oil uses a binary packaging design, which is a multiprotection material packaging system formed by valves, multilayer vacuum bags, and aluminum cans, which are explosion-proof and fireproof, while preventing the leakage of oil (Figure 5). Because of the system’s opacity and good sealing performance, the product’s exposure to air and ultraviolet rays is reduced, hindering the oil’s deterioration. An inert gas pressure container can be proposed to improve the oxidative stability as an alternative to microencapsulation, which also could improve the quality and stability of unsaturated SLs. This will allow improving the atomization efficiency and enhancing the sealing effect of spray oil, in addition to preventing exposure to oxygen and prolonging the shelf-life.

### 4.4. Emulsion

An emulsion is made up of two incompatible phases (generally oil and water), wherein one of the spherical droplets is dispersed in the other phase. According to the difference between the continuous phase and dispersed phase, emulsions can be divided into O/W and W/O. At present, emulsion research mainly involves microemulsions, nanoemulsions, Pickering emulsions, high-internal-phase emulsions, etc. The application of unsaturated SLs and other functional bioactives in the food industry is limited due to their inherent properties; for example, carotenoids, polyphenols, unsaturated fatty acids, etc. are often poorly soluble in water, sensitive to oxygen, and easy to degrade under light and heat conditions. In recent years, utilizing emulsion systems to carry functional bioactives via SL delivery systems has emerged as an attractive approach with fascinating prospects to improve their biological efficacy (Figure 6).

It was found that microemulsions, nanoemulsions, and Pickering emulsions can be prepared to load DHA/EPA, regulate their release in the gastrointestinal tract, protect them from adverse environmental effects, and improve their solubility and bioavailability [131]. The preparation of these emulsions usually requires the use of artificial surfactants, but the use of synthetic ingredients may lead to environmental damage and adverse health effects on consumers; therefore, it is necessary to design and select green, healthy natural surfactants to prepare emulsions. Liu and Li [132,133] found that medium- to long-chain DAG can be used to produce Pickering emulsions and high-internal-phase emulsions, thereby improving the stability of the emulsion, which enables the use of DAG emulsions in the preparation of low-calorie margarine, spreadable sauce, and cosmetics. Guo et al. [134] recently used an MLCT-based emulsion system loaded with vitamin D to perform in vitro and in vivo digestion experiments, which demonstrated the improved bioavailability of vitamin D. Studies have demonstrated the possibility of preparing emulsions with diglycerol as an emulsifier and preparing emulsions loaded with easily oxidized structural oils. Currently, the application of SLs in emulsion systems is attracting increasing attention. In the future, researchers could explore the properties of SLs in emulsion systems for the transport of nutrients, so as to improve their intake, which is conducive to achieving a healthy diet.

## 5. Conclusions

Enzymatic synthesis shows great advantages in the preparation of SLs. The enzyme reactor is the core device of an enzyme-catalyzed reaction, and selecting the appropriate reactor is beneficial to the performance of the reaction. Further research should be carried out to achieve the large-scale application of PBR and CBR in industry. Novel oils such as MLCT, DAG, EPA- and DHA-enriched TAG, human milk fat substitutes, and EPG can be prepared through esterification, transesterification, acidolysis, and their combinations. In the future, it will be crucial to explore ways to extend the shelf-life of easily oxidized and unstable structural functional lipids by applying SLs to some food systems such as powdered oil, plastic fat, and emulsions, while also improving their functional characteristics and maximizing their nutritional properties. The application of high-melting-point DAG as a novel base stock has large potential value in the production of specialty fats with low saturated fat content, high stability, and decent operability. As compared to other oils with a higher melting point, the utilization of MLCT with a lower melting point, as well as DHA- and EPA-enriched TAG, represents a new research pathway for the development of novel carrier systems essential for the delivery of healthy functional elements in the future.

## Figures and Tables

**Figure 1 foods-11-02400-f001:**
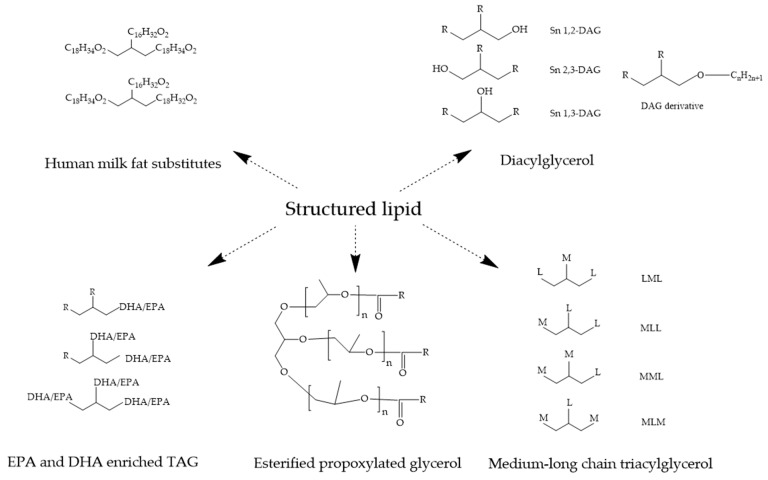
The structure of five structured lipids. (R, M, L, C_18_C_34_O_2_, C_16_H_32_O_2_, and C_18_H_32_O_2_ represent fatty acids, medium-chain fatty acids, long-chain fatty acids, oleic acid, palmitic acid, and linoleic acid, respectively).

**Figure 2 foods-11-02400-f002:**
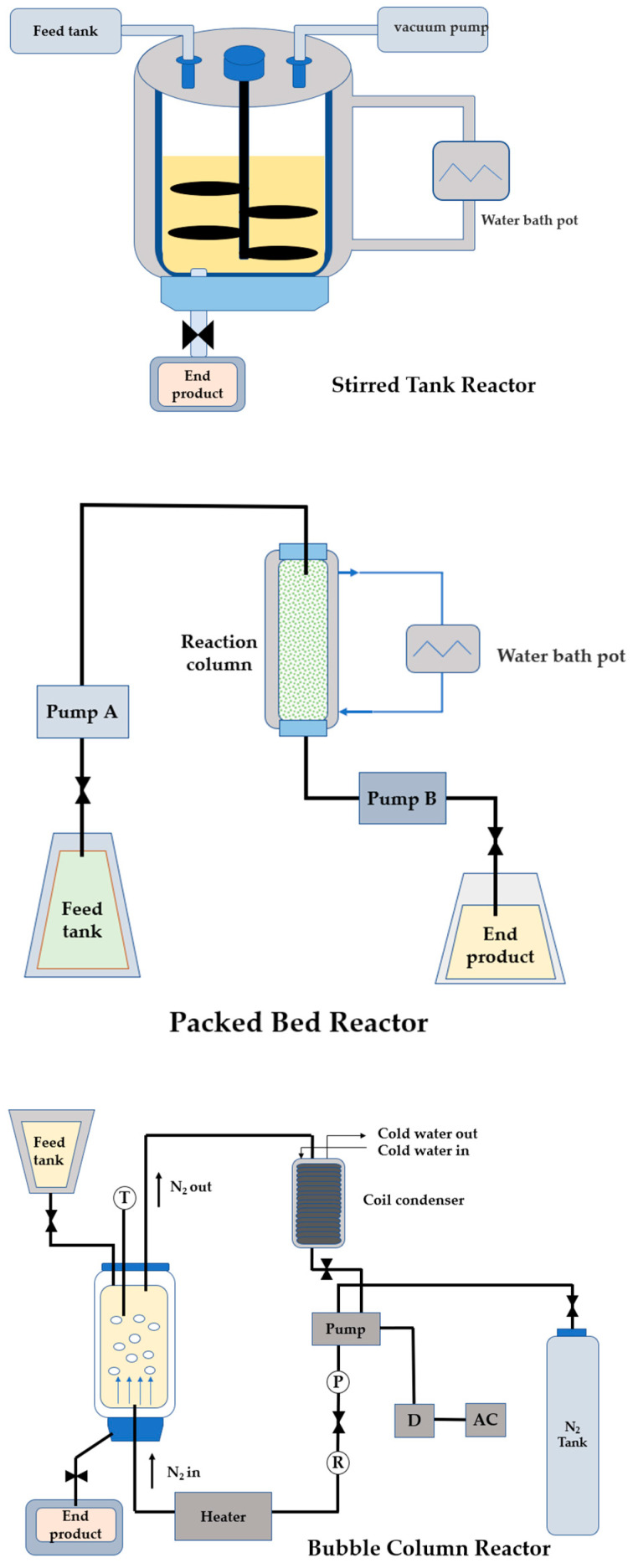
The Schematic diagram of several common enzyme reactors.

**Figure 4 foods-11-02400-f004:**
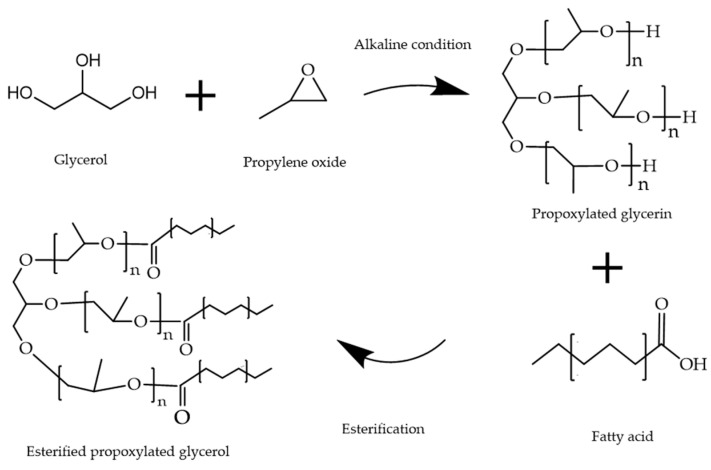
The systhesis mechanism of esterified propoxylated glycerol (EPG).

**Figure 5 foods-11-02400-f005:**
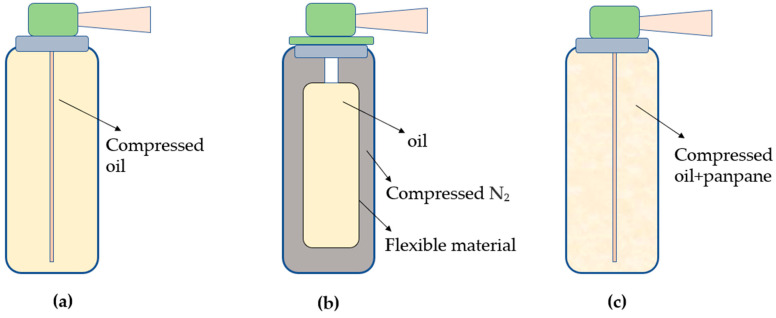
The schematic diagram of several spray oils: (**a**) is traditionally used to generate spray oil by compressing oil, (**b**) is used to form spray oil by compressing inert gas, and (**c**) is used to generate spray oil by compressing propane and oil together.

**Figure 6 foods-11-02400-f006:**
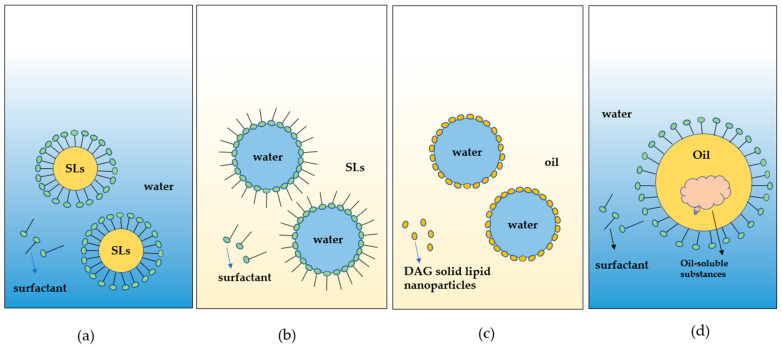
Schematic diagram SLs-based emulsions. (**a**) Oil-in-water emulsion; (**b**) water-in-oil emulsion; (**c**) Pickering emulsion formed from DAG-based nanoparticles as surfactants; (**d**) oil-in-water emulsion loaded with oil-soluble substances.

**Table 1 foods-11-02400-t001:** The advantages and disadvantages of the use of different methods in SLs production.

	Enzymatic Method	Chemical Method	Genetic Method
Reaction condition	Mild; Efficiency; Green	Strict to catalyst and substrates; Flexible; Stability; Chemicals	Strong pertinence; Complex process; Long cycle
Production cost (Energy consumption, catalyst reuse ability)	High lipase cost; Continuous; Low energy consumption	Low catalyst cost; Batch; High energy request	High breeding cost
Production efficiency	High yield; Specificity	Easy to degrade; Random	High yield; Specificity
Production security	High security in solvent-free system	Chemical and metal issues	Potential gene security issues
References	[4,8,9,10,11,12]	[4,13,14]	[4,15,16]

**Table 2 foods-11-02400-t002:** Current enzyme reactors for SLs production.

Equipment	Substrates	Target Products	Conditions	Advantages	References
Orbital shaker	Fully hydrogenated soybean oil, caprylic acid	MLCT	PyLip enzyme acidolysis at 60 °C for 1 h, hexane	Substrates mixed evenly; Controllable reaction conditions	[28]
Rotary shaker	Flax oil, single cell oils, caprylic acid	MLCT	Immobilized TLL lipase acidolysis at 50 °C for 24 h, solvent-free system	Controllable reaction conditions; Low energy consumption	[29]
Stirrer	Extra virgin olive oil, soybean oil, fully hydrogenatedcrambe oil	Behenic acid enriched SL	Lipozyme TL IM interesterification at 60 °C for 4 h, solvent-free system	Controllable reaction conditions; Low energy consumption	[30]
Shaker	Mutton tallow, hemp oil	DAG	Immobead 150 lipase interesterification at 60 °C for 6 h	Substrate mixed evenly, Controllable reaction conditions	[31]
Batch reactor	Grapeseed oil, capric acid, caprylic acid	MLCT	Lipozyme RM IM^®^ lipase acidolysis at 45 °C for 24 h, solvent-free system	Controllable reaction conditions	[32]
Magnetic stirrer	High oleic sun-flower oil, fully hydrogenated Crambe abyssinica oil	Behenic acid enriched SL	Lipozyme TL IM interesterification at 70 °C for 3 h, solvent-free system	Low cost; Controllable reaction conditions	[33]
Bubble column reactor	Caprylic acid, capric acid	High Purity Medium Chain DAG	Novozym 435 esterification at 60 °C for 30 min, solvent-free system	Mild; Efficiency; Durable catalyst	[27]
Bubble column reactor	Palm oil deodorizer distillate, oleic acid, glycerol	DAG	Lipozyme 435 esterification at 60 °C for 30 min, solvent-free system	Mild; Efficiency; Durable catalyst	[34]
Packed bed reactor	Palm olein, fully hydrogenated palm oil, palm kernel oil	Cocoa butter substitutes	Lipozyme TL IM interesterification at 65 °C with feed flow rate of 70 mL/min, solvent-free system	Mild; Efficiency; Durable catalyst	[35]
Magnetic stirrer	MAG, caprylic acid	DAG	Novozyme 435 esterification at 65 °C for 30 min, solvent-free system	Low cost; Controllable reaction conditions	[36]

**Table 3 foods-11-02400-t003:** Current enzymatic synthesis methods to produce MLCT.

Substrates	Type and System of Reaction	Optimal Conditions	Characteristics	References
Caprylic acid, capric acid, oleic acid, glycerol	Vacuum and solvent-free system, esterification	Novozym 435 esterification at 90 °C for 12.37 h	MLCT content, Enzyme activity	[6]
Arachidonic acid single cell oils, MCTs	Solvent-free system, transesterification	Lipozyme 435 transesterification at 90 °C for 3 h.	Fatty acid and TAG composition, Melting and crystallization behavior	[37]
Glycerol, capric acid, oleic acid	Vacuum and solvent-free system, esterification	Lipozyme RM IM esterification at 70 °C for 14 h	MLCT content	[40]
Canola oil, caprylic acid	Solvent-free system, acidolysis	Lipozyme RM IM acidolysis at 50–60 °C for 15 h	Fatty acid and TAG composition, Refractive index, Melting profile	[41]
Cinnamomum camphora seed oil, camellia oil	Solvent-free system, transesterification	Lipozyme RM IM transesterification at 60 °C for 3 h	Fatty acid and TAG composition, Tocopherol analysis	[42]
Flaxseed oil, tricaprylin	Organic solvent system, transesterification	Lipozyme TL IM transesterification at 41.49–50.00 °C for 4.00–4.01 h	Bioconversion yield	[43]
Hydrogenated soybean oil, rice bran oil, coconut oil	Solvent-free system, transesterification	Lipozyme TL IM transesterification at 65 °C for 24 h	Fatty acid and TAG composition, Polymorphism, Crystal Morphology, Analysis of Tocopherols and Phytosterols	[44]
Soybean oil, MCT	Solvent-free system, transesterification	Lipozyme TL IM transesterification at 55 °C for 30–40 min	Fatty acid and TAG composition, DAG content, Acyl migration	[45]
Catfish oil, basa catfish oil	Solvent-free system, transesterification	NS 40086 lipase transesterification at 60 °C for 3 h	Fatty acid and TAG composition,	[46]
Microbial oil, MCT containing 99% of caprylic acid	Solvent-free system, transesterification	NS 40086 lipase transesterification at 60 °C for 8 h	Fatty acid and TAG composition	[47]

**Table 4 foods-11-02400-t004:** Current enzymatic synthesis methods to produce DAG.

Substrates	Type and System of Reaction	Conditions	Characteristics	References
Menhaden oil	Solvent-free system, glycerolysis	Novozym 435 glycerolysis at 70 °C for 24 h	DAG content, Enzyme activity, Positional Analysis of TAG, Fatty acid composition	[56]
Rice bran oil	Solvent-free system, glycerolysis	C.antarctica lipase glycerolysis at 70 °C for 24 h	MAG and DAG content, Particle size	[63]
Olive oil	Solvent-free system, glycerolysis	Novozym 435 glycerolysis at 70 °C for 4 h	Fatty acid composition, Enzyme activity	[64]
Oleic acid	Solvent-free system, esterification	Lecitase^®^ Ultra esterification at 40 °C for 1.5 h	Acylglycerols compositions, Enzyme activity	[65]
Lauric acid	Solvent-free system, esterification	Lipozyme RM IM esterification at 50 °C for 3 h	DAG content, Reusability of lipase	[66]
Rapeseed oil, MAG, oleic acid	Solvent-free system, esterification	Immobilized lipase EC3.1.1.3 at 60 °C for 6 h	Acid value, MAG and DAG content	[67]
Soybean oil	Solvent-free system, glycerolysis	Immobilized RML glycerolysis at 60 °C for 24 h	DAG content, Enzyme activity, TAG conversion	[68]
Soybean oil	Solvent-free system, glycerolysis	Immobilized TLL glycerolysis at 60 °C for 12 h,	MAG, DAG, and TAG content, Enzyme activity	[69]
Soybean oil	Solvent-free system, glycerolysis	Immobilisation lecitase^®^ ultra glycerolysis at 45 °C for 12 h	TAG composition, DAG content	[70]
Short- and medium-chain fatty acid ethyl esters	Solvent-free system, transesterification	Novozym 435 transesterification at 65 °C for 24 h, Lipozyme RM IM transesterification at 65 °C for 32 h.	MAG, DAG, and TAG content	[71]

**Table 5 foods-11-02400-t005:** Current enzymatic synthesis methods to produce EPA- and DHA-enriched products.

Substrates	Type of Reaction	Conditions	References
Fish oil	Two-stage enzymatic refining process: hydrolysis, transesterification	400SD hydrolysis at 35 °C for 10 h, sodium phosphate buffer; Novozym 435 transesterification at 60 °C for 6 h, solvent-free system	[88]
Codfish oil	Hydrolysis	OUC-Lipase 6 hydrolysis at 40 °C, for 36 h, Tris-HCl buffer	[89]
Camelina sativa oil, omega-3 fatty acid ethyl esters	Two-step reaction: ethanolysis, transesterification	Lipozyme TL IM ethanolysis for 1 h, ethanol; Novozym 435 transesterification at 35 °C for 4 h, solvent-free system	[90]
Glycerol, DHA/EPA-rich ethyl esters	Two-step enzymatic reaction: transesterification, ethanolysis	Novozym 435 transesterification at 60 °C for 24 h, solvent-free system; Immobilized SMG1-F278N ethanolysis at 30 °C for 96 h, *n*-hexane, ethanol	[91]
Glycerol, n-3 PUFA	Esterification	Novozym 435 at 50 °C for 50 h, deep eutectic solvents	[92]
Arctic cod liver oil	Alternate winterization and enzymatic interesterification	Alternate winterization at −80 °C for 24 h, acetone; Lipozyme TL IM interesterification at 40 °C for 2.5 h, solvent-free system	[93]
Microalgae oil and oleic acid	Acidolysis	Lipase RM IM acidolysis at 65 °C for 6 h, solvent-free system	[94]
Schizochytrium sp. biomass	Ethanolysis	Four liquid formulated enzymes CALA, PLA, RM and TL ethanolysis at 35 °C for 96 h, ethanol	[95]
Refined sardine oil, glycerol and tertpentanol	Glycerolysis	Lipozyme 435 glycerolysis at 50 °C for 2 h, *tert*-Pentanol	[96]
Salmon frame bone oil	Alcoholysis, esterification	Novozym 435 alcoholysis at 37 °C for 3 h, ethanol; Lipozyme RM IM esterification at 55 °C for 48 h, solvent-free system	[97]
Caprylic acid, A. limacinum SR21 oil	Acidolysis	Lipozyme TL IM acidolysis at 37 °C for 30.4 h, hexane	[98]
Microbial oil, medium-chain fatty acids	Acidolysis	Lipozyme RM IM acidolysis at 55 °C for 6 h, solvent-free system	[99]

**Table 6 foods-11-02400-t006:** Current enzymatic synthesis methods to produce human milk fat substitute SLs.

Equipment	Substrates	Conditions	Type of Reaction	References
Stirrer	Leaf lard, camellia oil fatty acids	Fractionation at 60 °C for 20 min followed by 34 °C for 10 h; Lipozyme RM IM acidolysis at 45 °C for 6 h, solvent-free system	Fractionation and acidolysis	[103]
Batch reactor	Palm stearin, oleic acid	NS40086 lipase acidolysis at 60 °C for 4 h, hexane, solvent-free system	Acidolysis	[105]
Magnetic stirrer	Palm stearin, oleic acid, linoleic acid	NS40086 lipase acidolysis at 60 °C for 4 h, solvent-free system	Acidolysis	[106]
Stirrer	Rapeseed oil, tripalmitin	Candida cylindracea lipase hydrolysis at 40 °C for 2 h, Tris-HCl buffer; Novozym 40086 acidolysis at 40 °C for 2 h, n-hexane	Hydrolysis and acidolysis	[111]
Magnetic stirrer	Soy oil, palm kernel stearin, palm stearin, oleic acid, linoleic acid	Lipozyme TL IM interesterification at 60 °C for 5 h, solvent free system; Lipozyme RM IM interesterification at 56 °C for 7 h, solvent free system	Interesterification	[112]
Stirred tank reactor and continuous PBR	Palm stearin, oleic acid	Stirred tank reactor: Immobilized AOL lipase at 65 °C for 1.5 h, solvent-free system PBR: Immobilized AOL lipase at 62.09 °C for 3 h, solvent-free system	Acidolysis	[113]

**Table 7 foods-11-02400-t007:** Approaches for preparation of powdered oil products.

Core Materials	Wall Materials	Preparation Methods	References
DHA-enriched fish oil	Zein	Electrospraying assisted by pressurized gas	[118]
Roasted coffee oil	Starch, gelatin, gum arabic	Spray drying and complex coacervation	[119]
Flaxseed oil	Lentil protein, maltodextrin	Freeze-drying	[121]
Cinnamon oil	Maltodextrin, gum arabic	Spray drying	[122]
Algal oil	Soy protein isolate, chitosan	Complex coacervation	[123]
Tilapia oil	Trehalose, gelatin, sucrose, xanthan	Spray drying	[124]
Coffee oil	Mesquite gum, octenyl succinic anhydride modified starches	Spray drying	[125]
Flaxseed oil	Maltodextrin, gum Arabic, whey protein concentrate, modified starch	Spray drying	[126]
Fish oil	Skim milk powder, whey protein concentrate, whey protein isolate, milk protein concentrate, sodium caseinate	Spray drying	[127]
Fish oil	Skim milk powder	Spray drying	[128]

## Data Availability

The data presented in this study are available on request from the corresponding author.

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
