# Peer review of "Future of Structured Lipids: Enzymatic Synthesis and Their New Applications in Food Systems"

_foods, 2022, doi:10.3390/foods11162400_

Round 1

Reviewer 1 Report

Reviewer comments and suggestions

Manuscript ID: foods-1841831

With this review paper, authors explore the enzymatic production of structured lipids and news applications of these compounds in food.  

Papers addressing this issue are important because nowadays the discovery of new ingredients and methodologies that contribute to the food industry and consumers’ health, are essential for increasing economy and life quality.

In my opinion,  some information should be completed and corrected.

Comments on the whole text: 

-  - Please check the punctuation and spaces in the text.

- - Check the text to the end of the line do not leave alone digits or values separated from the unit.

- - Check the text in terms of language, so as to constantly keep one time.

-The importance of this review must be strongly supported as compared with previous publications, showing the added value of this article, since this subject has already been addressed by other authors.

- - The importance of the use of structured lipids in food, to increase the consumers’ acceptability and health must be emphasized.

- - This paper must clearly state the originality of this review and the impact of this study on the functional food industry. Why has this theme steadily been a research hotspot?

- - The advantages and disadvantages of the use of enzymatic synthesis over chemical and genetic methods, in the preparation of structured lipids must be explored. A Table should be presented containing this information.

-  - Table 5 needs to be revised as it is very confusing.

- - The quality of the figures should be improved, and Figure 5-c) must be corrected.

- - When the authors approach the applications of structured lipids in future food systems, the authors should refer to the existing/needs of legislation.

- - Authors should clarify what they mean by “….developing new food systems”.

- - Ensure that all references are the most recent and relevant to the arguments in the paper.

Final comments and considerations: It deserves to be published at Foods after the suggestions and corrections listed above are amended.

Reviewer 2 Report

Future of structured lipids: Enzymatic synthesis and their new applications in food system.

This is an actual and interesting topic, but actually, the manuscript is not in a comprehensive state, lacks arguments and discussion accuracy, and in the case of future perspectives proposed, they are sometimes not mentioned or are out of context with the objectives of the review work. In addition, the manuscript requires a deep English grammar revision and correction. Following are examples of the main issues: 

1.     Line 34-37. This sentence gives general information, but its redaction is not linked with the above or next sentence. I suggest associating it adequately or eliminating it for better fluidity of the writing.

2.     Line 38. The fat substitutes have been presented as alternatives to control excessive fat consumption, this is not a result of fat consumption. Please, be careful with the order of the ideas you are redacting or when you traduce to the English language.

3.     Line 42. Again, “As a result”, this grammar connection is not adequate here. It changes the sense of the context. Please, correct it.

4.     In general, the first paragraph must be reframed for better understanding.

5.     Line 49-50. These two paragraphs are in the same way, they should be together. After coma (line 50), there is a different idea this should be separate. You must attend it and enhance the redaction. The whole manuscript is full of cases like this.

6.     Line 53-54. Grammar issues, this manuscript needs a deep grammar revision and correction.

7.     Lines 193-197. Between these lines, there are two sentences repeated and it's not understood. The complete paragraph is confusing.

8.     Line 222. Here, this sentence should start: At the difference of OPO, OPL contains linoleic acid...." because as is written is understood that only is "composed by linoleic acid... Please, read it again and try to enhance the redaction.

9.     Line 227- 229. Is it really cheaper for the enzymatic process to produce OPO or OPL for infant formulas? Please, add the references to support it and discuss your overview.

10.  Line 239. Section 2.5. What about the future trends of EPG? The actual applications sound really interesting for the food industry. This section lacks your discussion about EPG opportunities in the production and applications fields.

11.  Line 262. Section 3. What is the relevance of including at this point a section with the reaction systems for SL? It would be more interesting if you discuss them in each of the previous processes when it mentions the preparation and later the chemical reactions for the production of the SL. Another option is to distribute the information in Table 5 in the respective section that describes the specific SL. Or also, this section could be before all the SL types, in order to follow the production processes steps.

12.  Line 268-270. Despite all the advantages mentioned above in the different SL production processes, is this process feasible and well used today? It is necessary to discuss the relevance of including this information in the section.

13.  Line 309. Section 4. I suggest changing the title to "Future applications of SL in food systems" because the food already exists but the SL addition in them would be the innovation... But when I read all the section, I can see that most of the systems mentioned are not obtained by enzymatic synthesis, and all of the "future" products already exists, so the section is out of the context of the objectives of this review.

14.  Line 335. Section 4.2. In this section is not included any information about the uses or proposed uses of SL. The information is out of context with the topic of this manuscript.

15.  And the future trends? The applications mentioned already exist. 

Reviewer 3 Report

The manuscript provides an interesting review of the production of structured oils by enzymatic synthesis. However, some aspects of the review need to be included to improve the current state of the work. The comments are given in detail below:

Page 2, lines 47-48. The authors mention that there are three methods to produce structured oils. Therefore, it is suggested to include a Table with the advantages and disadvantages of the enzymatic synthesis method compared to the chemical (widely used) and genetic modification methods.

Page 20, line 317. In this section, the authors should point out that there are currently some studies on structured oils in powder form using the chemical method (indirect oleogelation) to structure the oil.  However, there are no studies on structuring powdered oils using enzymatic synthesis.

Page 25, lines 403-404. The authors indicate few studies on preparing emulsion systems using structured oils. However, there is quite a lot of information on the subject. Therefore, it is suggested that this paragraph be revised.

Round 2

Reviewer 1 Report

The authors replied to my comments and they have provided a new and improved version of the paper.

Reviewer 2 Report

I have revised the manuscript and I can see the author's well-attended observations and that they have made consciously almost all the corrections suggested. So, in my opinion, this work can be considered for publication in the Foods journal.